# Temperature, Humidity and Regional Prevalence of Dry Eye Disease in Argentina

**DOI:** 10.3390/medicina61122226

**Published:** 2025-12-17

**Authors:** María C. Marini, Belén Liviero, Rodrigo M. Torres, Jeremías G. Galletti, Gustavo Galperin, Monica Alves, Jesús Merayo-Lloves

**Affiliations:** 1Cornea and Ocular Surface Department, Hospital de Alta Complejidad El Cruce, Florencio Varela-Hospital Británico, Buenos Aires B1888, Argentina; mariaceciliamarini@gmail.com; 2Cornea and Ocular Surface Department, Hospital Británico, Buenos Aires City C1275AGB, Argentina; 3Ocular Surface Department, Humana Centro Médico, Córdoba X5009LDF, Argentina; belenliviero@gmail.com; 4R.O.M.A.T. Creator Center, Colonia Avellaneda E3107, Argentina; 5Institute of Experimental Medicine, CONICET-National Academy of Medicine of Buenos Aires, Buenos Aires C1425AUM, Argentina; jeremiasg@gmx.net; 6Hospital Oftalmológico Pedro Lagleyze, Ciudad de Buenos Aires, Buenos Aires C1425CLA, Argentina; galperingustavo@gmail.com; 7Department of Ophthalmology and Otorhinolaryngology, University of Campinas, Campinas 13083-970, Brazil; monicalves@me.com; 8Instituto Oftalmológico Fernandez Vega, 33012 Oviedo, Spain

**Keywords:** dry eye disease, epidemiology, climate, relative humidity, temperature, Argentina, ocular surface

## Abstract

*Background and Objectives*: To evaluate whether regional climatic factors, particularly relative humidity (RH) and temperature, are associated with geographic variation in dry eye disease (DED) prevalence in Argentina. *Materials and Methods*: A secondary analysis of a nationwide voluntary online cross-sectional survey of DED in Argentina (April 2022; non-probability sample; *n* = 10,812) was performed. Participants were assigned to one of five standard geopolitical regions. DED prevalence (%) was estimated at regional levels. Monthly mean, maximum, and minimum temperature and RH for April 2022 were obtained from the National Meteorological Service; for each province/region, data corresponded to the principal synoptic station in the provincial capital. Because the survey and climate data were restricted to April, seasonal effects could not be evaluated. Region-level analyses of climate–DED patterns were prespecified as exploratory and descriptive given the small number of regional units (*n* = 5). Weighted descriptive models were used to compare RH–DED patterns while accounting for regional age and sex distributions. *Results*: Regional DED prevalence ranged from 37.7% (Northwest) to 49.8% (Cuyo). RH showed a clear inverse pattern with regional DED prevalence, whereas temperature showed no consistent relationship. Weighted descriptive models showed a similar qualitative RH–DED pattern, although residual confounding cannot be excluded. *Conclusions*: DED prevalence in Argentina shows geographic variability and a descriptive inverse pattern with ambient RH. These ecological findings are hypothesis-generating and must be interpreted cautiously due to sampling bias. Future seasonal, longitudinal, or individual-level studies are needed to confirm and quantify these preliminary signals.

## 1. Introduction

Dry eye disease (DED) is a multifactorial and increasingly prevalent global health problem. The recent TFOS DEWS III (2025) defined DED as a disease characterized by loss of homeostasis of the tear film and ocular surface, involving tear film instability, hyperosmolarity, inflammation, epithelial damage, and neurosensory dysfunction [1]. Environmental determinants have been repeatedly highlighted as relevant in its pathophysiology.

The ocular surface is constantly exposed to ambient conditions; changes in those conditions can destabilize tear film equilibrium [2]. In the context of climate variability, meteorological exposures have gained relevance as potential risk factors for DED [2,3,4,5,6,7]. Relative humidity (RH) governs evaporative rate: when below saturation, water loss from the tear film increases, particularly with lipid layer abnormalities [8]. A growing body of international research, from North America, Europe, Africa, and East Asia, demonstrates that climatic and environmental stressors affect ocular surface homeostasis, with low ambient humidity, high wind exposure, extreme temperatures, and increased air pollution being associated with tear film instability, shorter tear breakup time (TBUT), and greater symptom burden [3,4,5,6,7,9,10,11,12,13,14,15].

Global estimates of DED vary widely (5–50%) depending on population and diagnostic criteria [16,17,18,19,20,21,22,23,24]. Beyond demographic and methodological factors, geographic and climatic heterogeneity likely contributes to these differences [2,3,4,5,6,7]. From a physiological standpoint, RH modulates evaporative loss, while temperature gradients can alter lipid layer viscosity and blink dynamics. Wind increases convective evaporation, and altitude reduces atmospheric pressure, both of which favor tear film thinning. These mechanisms provide biological plausibility for regional and seasonal variation in DED prevalence.

Argentina (2,791,810 km^2^; 45.9 million inhabitants) spans diverse latitudes and climates, subtropical in the north, temperate in the center, and subpolar in the south, and is commonly divided into five geopolitical regions for analysis [25]. This heterogeneity arises from marked latitudinal gradients (≈34 degrees from north to south), major altitudinal variation (from sea level to over 6000 m in the Andes), and contrasting maritime versus continental influences. As a result, RH and temperature can differ substantially between regions, making the country a suitable setting to explore climate–DED relationships.

In 2024, a nationwide online survey reported an overall DED prevalence of 42% in Argentina [26]. In that same cohort (*N* = 10,812), female sex and older age emerged as the principal individual-level correlates of DED in multivariable models. Building on those findings, the present study examines whether regional climate, especially RH and temperature, contributes to the geographic patterning of DED across Argentina.

## 2. Materials and Methods

A secondary analysis was performed using the database from the first nationwide epidemiological study on DED in Argentina [26]. This cross-sectional survey was conducted in April 2022 through an anonymous, voluntary electronic questionnaire administered to the general population. The non-probability sample included individuals aged ≥12 years residing in the 23 provinces of Argentina and the Autonomous City of Buenos Aires. Because sampling was non-probabilistic, prevalence estimates and regional differences are descriptive of the surveyed sample and are not population-representative. The study was approved by the Ethics Committee of Hospital El Cruce and conducted in collaboration with the Argentine Council of Ophthalmology (CAO) and the Argentine Society of Ocular Surface (SASO), in accordance with the Declaration of Helsinki.

The questionnaire was based on the Women’s Health Study (WHS) Dry Eye Questionnaire [27]. DED was defined as the presence of severe symptoms (constant or frequent dryness or irritation) identified through the questionnaire, or a positive response regarding a previous ophthalmologist-confirmed diagnosis of DED. Additional questions were included to assess demographic characteristics, lifestyle, general medical history, medication use, and ocular conditions [26]. Given that the case definition combined severe symptoms with prior ophthalmologist-confirmed diagnosis, differential access to eye care and awareness across regions may influence case ascertainment.

For the present analysis, participants were assigned to province and region according to residence. DED prevalence (%) was estimated overall and by province/region. Climatic exposures, monthly mean, maximum, and minimum temperature (°C) and monthly mean relative humidity (RH, %) for April 2022, were obtained from the National Meteorological Service and linked by province. For each province, we used the principal synoptic station in the provincial capital (the Service’s official climatological reference). Provincial proportions were explored but not reported, as several province-level estimates were unstable due to uneven sample sizes and the ecological nature of capital-station exposures; results are therefore presented at the prespecified regional level to reduce misinterpretation risk. Values are simple monthly means derived from daily observations; no weighting, altitude adjustment, or urban–rural correction was applied, as the aim was to characterize broad regional profiles rather than micro-environmental variability. Because the original survey was conducted exclusively in April, we used April data to maintain temporal alignment; seasonal or multi-month exposures were not evaluable. These monthly, capital-based measures constitute a coarse ecological proxy and may not capture short-term extremes or within-province heterogeneity.

Regarding statistical analysis, qualitative variables were summarized as percentages and quantitative variables as mean ± SD or median (IQR), as appropriate. Because sampling was non-probabilistic, prevalence estimates and regional differences reflect the surveyed sample and are not population-representative. For provincial and regional DED proportions, 95% confidence intervals were computed using the Wilson method. Between-group comparisons used Chi-square tests and one-way ANOVA or Kruskal–Wallis, as indicated. Given that demographic covariates (age and sex distribution) were stable only at the regional level, and that provincial-level data did not allow adjusted analyses with sufficient robustness, climate–DED associations were evaluated using prespecified exploratory analyses at the level of the five geographic regions. These ecological analyses were strictly descriptive due to the small number of regional units (*n* = 5). Region-level relationships between DED prevalence (%) and monthly mean RH and temperature were examined using Pearson correlations and weighted correlations (weights equal to regional sample sizes). Additionally, weighted least-squares (WLS) regressions were fitted with DED% as the dependent variable and RH as the main independent variable in three parsimonious models (DED% ~ RH; DED% ~ RH + mean age; DED% ~ RH + proportion of women), using regional sample sizes as weights. These models were intended solely to describe the direction and relative magnitude of potential climate–DED patterns and to explore whether age or sex acted as confounders. No *p*-values, confidence intervals, or inferential interpretations were applied to these ecological models, as regression estimates with *n* = 5 are statistically unstable and cannot support conventional hypothesis testing. All ecological results should therefore be interpreted as descriptive only. Two-tailed *p* < 0.05 denoted statistical significance for non-ecological comparisons. Analyses were performed in R (v4.4.2).

## 3. Results

A total of 10,812 questionnaires were collected, as previously reported [19]. For the present analysis, we focused on the geographic and climatic distribution of DED across Argentina.

### 3.1. Regional Prevalence of DED and Demographic Characteristics by Region

DED prevalence varied across the five Argentine regions (Table 1). The highest proportions were observed in Cuyo and the Pampas, whereas the Northwest and Northeast displayed comparatively lower values. Patagonia, although intermediate in percentage prevalence. We report regional prevalence as percentages in Figure 1. These regional differences illustrate a heterogeneous geographic pattern of DED across the country. Importantly, Cuyo, the region with the lowest measured RH (55.1%), showed the highest DED prevalence (49.8%), whereas the Northwest, which had the oldest age profile among respondents, exhibited the lowest prevalence (37.7%). These descriptive contrasts help contextualize how demographic and climatic gradients align geographically.

The proportion of women was consistently high across all regions, with only modest variation (Table 1). Mean age differed more notably: the Northwest showed the oldest age profile, whereas the Northeast had the youngest. These demographic patterns provide contextual information but do not mirror the geographic pattern of DED prevalence.

### 3.2. DED Prevalence, Climate Patterns, and Geographic Distribution

Marked climatic variability was observed across regions. The Northeast and Northwest had the highest RH values, while Cuyo showed the lowest. This climatic contrast parallels the descriptive pattern observed in prevalence, where the driest region (Cuyo) had the highest DED percentage, and the most humid regions showed comparatively lower values. Patagonia presented the lowest temperatures, while the Northeast registered the highest (Table 2; Figure 2). These gradients reflect the well-established climatic diversity of Argentina.

A visual comparison of regional DED prevalence with climatic measures revealed a notable inverse relationship between DED prevalence and RH: regions with lower humidity tended to show higher DED prevalence (Figure 2). In contrast, no consistent pattern emerged between DED prevalence and regional temperature (mean, maximum, or minimum). These patterns are described qualitatively only, as the five-region design precludes inferential interpretation.

Regional temperature (mean, maximum, and minimum) did not show an observable pattern in relation to DED prevalence. As illustrated in Figure 3, the regional distributions of DED overlapped across the temperature range, without a discernible gradient. Accordingly, temperature is presented descriptively, whereas Table 2 focuses on RH and demographic characteristics, which exhibited clearer geographic variation.

## 4. Discussion

In our previous nationwide epidemiological study of DED in Argentina, we reported a prevalence of 42% in a population of 10,812 participants [26]. Building on that foundation, the present work explored whether broad regional climatic characteristics, particularly RH and temperature, might help contextualize the geographic distribution of DED across the country. Given Argentina’s substantial latitudinal and climatic diversity, examining these environmental patterns may aid in understanding population-level differences in DED burden.

DED is the most frequent ocular surface disorder worldwide, and understanding its multifactorial nature is essential for its management [6,17]. Environmental exposures were highlighted as constant risk factors in the TFOS DEWS II report, alongside age, female sex, Asian race, meibomian gland dysfunction, Sjögren syndrome, connective tissue diseases, androgen deficiency, and modifiable factors such as digital screen use, contact lens wear, hormone replacement therapy, medications, and hematopoietic stem cell transplantation [6]. The recent TFOS DEWS III consensus reaffirmed the importance of environmental determinants in DED pathophysiology [1]. The ocular surface is continuously exposed to ambient conditions, and variations in humidity, temperature, wind, and altitude can influence the stability of the tear film and the regulation of ocular surface homeostasis [2,3,7,8].

To facilitate the interpretation of climatic influences within such a large territory as Argentina, we analyzed DED prevalence by geopolitical regions. Argentina spans latitudes that encompass subtropical climates in the north (Northeast and Northwest regions), temperate climates in the central areas (Pampas and Cuyo), and cold climates in the south (Patagonia). Humidity levels also vary with distance from the sea and topography: regions influenced by humid Atlantic winds (Northeast, Pampas, Patagonia) contrast with drier zones in Cuyo and parts of the Northwest where altitude and continentality predominate. These gradients are reflected in RH levels, which range from 55% in Cuyo to over 80% in parts of the Northeast. Consistent with established mechanisms, where low RH accelerates tear evaporation and destabilizes the tear film, DED prevalence was highest in the driest region (Cuyo) and lowest in the more humid Northwest. Conversely, temperature did not display a clear pattern across regions, and the distributional plots showed substantial overlap, suggesting that temperature was not a major driver of geographic variation during the month studied. In this context, the comparatively lower DED prevalence observed in the humid subtropical Northeast may be consistent with physiological mechanisms described in experimental studies. Higher ambient RH reduces the gradient for evaporative water loss, slows tear film thinning, and mitigates osmolar stress on the ocular surface. Although our ecological design precludes causal inference, these mechanisms offer a plausible explanation for why regions with higher RH—such as the Northeast and Northwest—display lower DED burden despite demographic differences.

The relationship between DED and climatic variables such as temperature and humidity has been widely studied [3,7,8]. Environmental conditions directly affect the ocular surface, which is permanently exposed [2]. According to the TFOS Lifestyle Report, factors such as temperature, humidity, wind speed, altitude, and ultraviolet radiation may disrupt ocular surface homeostasis and increase the risk of ocular surface diseases [1]. Low RH (<30–40%) and air currents (e.g., fans, air conditioning, heating) increase tear evaporation, reduce tear film stability, and exacerbate DED symptoms [1,8]. Experimental evidence shows that drier environments reduce tear film breakup time and increase osmolarity [28]. Both high temperatures with low humidity and cold, windy conditions are associated with adverse ocular surface effects [1,7,8,21].

Population-based evidence supports this association. In the DREAM study, participants living in Mediterranean climates had better tear film parameters, less corneal staining, and longer TBUT, correlating positively with higher temperature and humidity [29]. Similarly, a South Korean nationwide analysis found that low outdoor humidity was an independent risk factor for DED symptoms, with each 5% increase in humidity associated with a lower likelihood of reporting symptoms or diagnosis [30]. These findings align with our observation of an inverse correlation between ambient humidity and DED prevalence. In the descriptive ecological models, adjusting for age or sex did not materially change the direction of the RH–DED pattern; however, given the extremely small number of regional units (*n* = 5), these adjustments cannot be interpreted as evidence of lack of confounding. According to TFOS DEWS III, low humidity is a consistently evidenced risk factor for DED [1].

Evidence on indoor humidity is also relevant: multiple interventional studies have shown that manipulating indoor humidity improves signs and symptoms of DED, consistent with the fact that humans spend nearly 90% of their time indoors [31,32,33]. However, some reports suggest that excessively high indoor humidity may worsen symptoms, possibly by facilitating higher particulate concentrations or microbial growth [31,34].

Regarding temperature, results have been heterogeneous. An Indian study involving 500 participants found that high temperature and wind exposure increased DED prevalence [21]. In contrast, retrospective analyses in Asia showed that colder temperatures correlated with more severe symptoms and shorter TBUT [35]. In our cohort, no consistent temperature gradient was detectable; importantly, the study was conducted in April, a month without extreme seasonal variations in Argentina, and our figure plots show overlapping distributions across the temperature range (Figure 3).

The strength of this study lies in being the first report of its kind in South America using a large national dataset to explore geographic and climatic associations with DED. The single-month design allowed uniform meteorological comparisons. Nonetheless, several limitations merit emphasis. First, all ecological analyses rely on only five regions; therefore, correlations and regression coefficients must be interpreted strictly as descriptive patterns rather than inferential estimates, and no statistical significance can be assumed. Second, climatic exposures correspond exclusively to April 2022, the only month during which the nationwide survey was conducted. This necessarily limits the ability to evaluate seasonal variability or long-term environmental influences on this chronic condition. Argentina exhibits pronounced seasonal swings, and April represents a mild transitional month; thus, the humidity–DED patterns reported here should be viewed as exploratory signals rather than stable climatic associations. Moreover, monthly averaging can mask short-term episodes likely relevant to the ocular surface, such as very low relative humidity, strong winds, or temperature extremes, and capital-based measurements may not reflect within-province heterogeneity in large, geographically diverse regions. Accordingly, our exposure should be interpreted as a coarse ecological proxy. Future work using higher-frequency data (e.g., daily/hourly RH, wind), spatially resolved stations, or satellite/reanalysis products, and linking to indoor conditions, would better capture clinically meaningful variability.

A third limitation must be considered: the survey’s online, voluntary nature implies non-probability sampling with likely coverage, self-selection, and nonresponse biases; differential internet access and digital literacy may skew the sample toward younger, more educated, urban women, potentially inflating symptom-based prevalence and affecting regional contrasts. Accordingly, prevalence estimates and regional differences should be interpreted as sample-based rather than population-representative; no post-stratification weighting to census margins was applied. Also, our mixed case definition (severe symptoms or prior ophthalmologist-confirmed diagnosis) may influence the observed prevalence in regions with better access to eye care and higher awareness, independently of true disease burden. A symptoms-only definition could attenuate—but not eliminate—such differences, given regional variation in health-seeking behavior, digital literacy, and survey participation. Accordingly, regional contrasts should be interpreted with caution. In addition, the demographic imbalance observed across regions—most notably the disproportionately high proportion of women in Cuyo (≈85%)—suggests that sampling bias may contribute to the geographic gradients reported. Women are known to participate more frequently in online health surveys and to have a higher prevalence of DED worldwide, which may artificially elevate symptom-based prevalence in regions with over-representation of female respondents. Moreover, because sex is a strong determinant of DED and digital accessibility varies across Argentine regions, the unusually high proportion of women nationwide (≈80–86%) may itself distort regional comparisons. These factors must be taken into account when interpreting geographic differences, as part of the observed variation may reflect differential participation rather than true epidemiological differences.

Another consideration is that meteorological data reflect outdoor conditions and may not capture indoor environments, where individuals spend most of their time, potentially introducing nondifferential exposure misclassification. Indoor environmental conditions are strongly influenced by indoor climate-control systems (heating, ventilation, and air conditioning), which substantially alter humidity and airflow. As a result, the “humidity” exposure used here is an imperfect proxy for actual ocular surface stress. Such exposure misclassification is likely nondifferential and would typically bias climate–DED associations toward the null. This limitation also aligns with the ecological structure of the analysis: with only five regional units, even a true but modest ecological pattern may appear stronger or more stable than it actually is. Accordingly, the observed RH–DED relationship should be interpreted strictly as a descriptive, region-level pattern rather than an individual-level association.

Importantly, the use of a single month of meteorological data imposes a substantial constraint when studying a chronic condition such as DED. Correlating short-term outdoor humidity with long-term disease prevalence must therefore be interpreted with caution. Future investigations should incorporate multi-month or multi-year climatic datasets—including wind speed, ultraviolet radiation, and airborne pollutants—and ideally apply longitudinal, seasonal, or multilevel study designs linking individual-level outcomes to high-resolution environmental exposures.

Moreover, our regional models adjust only for age and sex; other determinants plausibly linked to both DED and ambient humidity, such as urbanization, occupational and screen-use patterns, contact lens wear, healthcare access, indoor climate control, air pollution, and wind exposure, were not available. As a result, regional RH may partly act as a proxy for these unmeasured behavioral and environmental factors, and the observed pattern cannot be attributed to humidity alone. Future studies using multilevel designs with individual-level covariates and granular environmental metrics (e.g., indoor RH, wind, pollutants) are needed to reduce residual confounding and test causality.

Despite these limitations, the descriptive geographic pattern observed suggests that environmental context, particularly regional differences in RH, could contribute to population-level variation in DED. These findings should be interpreted as hypothesis-generating rather than confirmatory. In line with recent evidence, including studies in contact lens wearers showing that low humidity and airflow exacerbate tear film instability and symptoms [36], our results reinforce the importance of considering both outdoor and indoor environments when evaluating risk factors. Although the present ecological findings cannot establish causality, they highlight potential avenues for future research into environmental prevention strategies. If longitudinal or individual-level studies confirm a causal link between environmental humidity and DED, this could inform discussions on whether environmental monitoring, indoor humidity optimization, airflow control, or season-specific educational initiatives might play a role in comprehensive public-health approaches. At present, however, such strategies remain speculative and should be explored only in the context of future confirmatory research.

## 5. Conclusions

In this nationwide ecological analysis, DED prevalence varied across Argentine regions and exhibited a consistent inverse pattern with ambient relative humidity. Because these observations are based on only five regional units, they remain descriptive and cannot support statistical inference or causal interpretation. Temperature did not demonstrate a discernible relationship with DED prevalence. These findings should therefore be regarded as hypothesis-generating. Future multi-seasonal, longitudinal, or individual-level studies are needed to confirm these preliminary signals and to evaluate finer-scale environmental exposures such as wind, airborne pollutants, and indoor microclimate. If confirmed, such evidence could contribute to climate-informed ophthalmic practice by refining risk stratification, guiding patient counseling, and supporting season-specific public-health messaging.

## Figures and Tables

**Figure 1 medicina-61-02226-f001:**
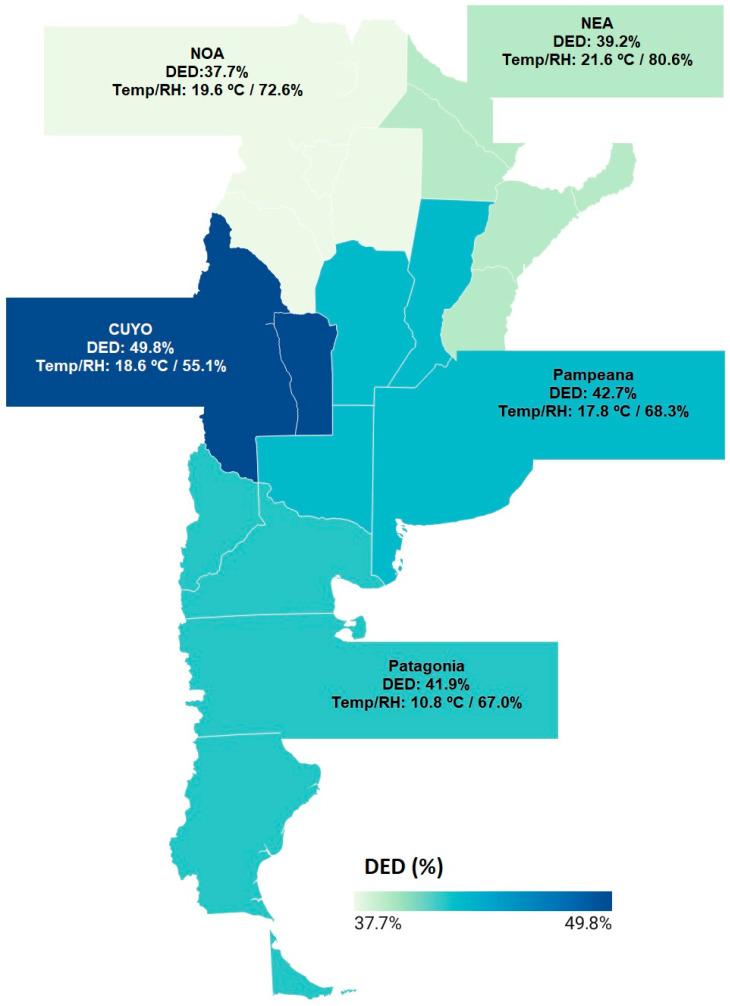
Regional map of Argentina showing DED prevalence (%). Shading encodes regional DED prevalence (range 37.7–49.8%). Colored boxes display the region name, regional DED (%), and monthly mean temperature/relative humidity. (DED: dry eye disease; NOA, Northwest Argentina; NEA, Northeast Argentina).

**Figure 2 medicina-61-02226-f002:**
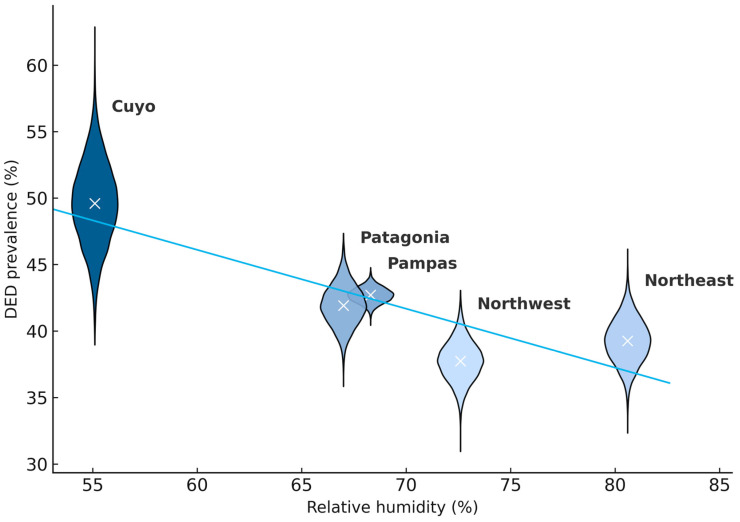
Correlation between regional DED prevalence (%) and monthly mean relative humidity (%).

**Figure 3 medicina-61-02226-f003:**
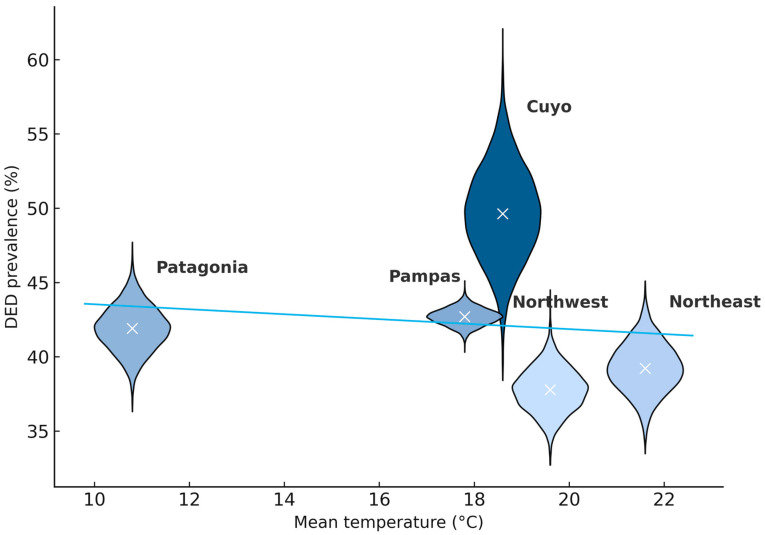
Correlation between regional DED prevalence (%) and temperature. DED: dry eye disease.

**Table 1 medicina-61-02226-t001:** Regional distribution of dry eye disease (DED) prevalence and monthly mean climatic variables (temperature and relative humidity) in Argentina.

Variable	Cuyo (*n* = 256)	Northeast (*n* = 836)	Northwest (*n* = 1060)	Pampas (*n* = 7457)	Patagonia (*n* = 1036)	*p*-Value
DED, *n* (%) [95% CI]	127 (49.8%) [95% CI: 43.5–55.7%]	328 (39.2%) [95% CI: 36.0–42.6%]	401 (37.7%) [95% CI: 35.0–40.8%]	3173 (42.7%) [95% CI: 41.4–43.7%]	435 (41.9%) [95% CI: 39.0–45.0%]	0.001
Women, %	85	81	80	85	86	0.022
Age, mean (SD)	43.7 (15)	41.9 (13)	55.5 (14)	46.5 (15)	45.2 (13)	<0.001
12–20 years, *n* (%)	6 (2.3)	9 (1.1)	9 (0.8)	39 (0.5)	13 (1.3)	0.001
21–40 years, *n* (%)	49 (19.1)	155 (18.5)	139 (13.1)	1120 (15.0)	147 (14.2)	<0.001
41–60 years, *n* (%)	52 (20.3)	136 (16.3)	199 (18.8)	1425 (19.1)	208 (20.1)	0.125
>60 years, *n* (%)	20 (7.8)	28 (3.3)	54 (5.1)	586 (7.9)	67 (6.5)	<0.001
Relative humidity, %	55.1	80.6	72.6	68.3	67.0	<0.001
Mean temperature, °C	18.6	21.6	19.6	17.8	10.8	<0.001
Max. temperature, °C	26.1	26.3	26.0	24.0	16.0	<0.001
Min. temperature, °C	11.1	17.0	13.3	11.5	5.5	<0.001

TP-values refer to overall differences across the five regions (Chi-square for categorical variables; one-way ANOVA or Kruskal–Wallis for continuous variables). DED 95% confidence intervals were computed using the Wilson method. Analyses are exploratory; no multiplicity adjustment was applied; (DED: dry eye disease).

**Table 2 medicina-61-02226-t002:** Weighted least-squares (WLS) descriptive models at the regional level with dry eye disease (DED) prevalence (%) as the outcome and relative humidity (RH, %) as the main exposure (*n* = 5 regions; weights = regional sample sizes).

Model	Covariates	RH Slope (pp per + 1% RH)	Interpretation
1	RH (%)	−0.409	Regions with higher RH tended to show lower DED prevalence.
2	RH (%), Mean age (years)	−0.407	Adjustment for age did not materially change the RH–DED pattern.
3	RH (%), Women (%)	−0.252	Adjustment for the proportion of women attenuated the RH–DED gradient.

These ecological models are strictly descriptive. With only five regional units, regression coefficients are not statistically stable and should not be interpreted inferentially. Values represent descriptive gradients only and do not imply causality. (DED: dry eye disease; RH: relative humidity; WLS: weighted least squares).

## Data Availability

The dataset with all the information through which the analysis has been carried out is available at (Zenodo): https://doi.org/10.5281/zenodo.10903549.

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
