# Peer review of "Temperature, Humidity and Regional Prevalence of Dry Eye Disease in Argentina"

_medicina, 2025, doi:10.3390/medicina61122226_

Round 1
Reviewer 1 Report
Comments and Suggestions for Authors
Thank you for the opportunity to review this manuscript.
Overall, this is a useful study with a clear message that is that lower relative humidity correlates with higher dry eye prevalence across Argentine regions. The study uses a very large sample size, ties the findings well to exisiting literature, writing is good and the visualizations are clean. However, there are several methodological weaknesses that limit the interpreatability and the strength of the conclusions. It stands good as a descriptive geographic report but it cannot support causal inferences whereas the authors overstate some associations.
Specific comments:
- The authors make correlations with n=5 regions, which are essentially uninterpretable statistically. The pearson r is meaningless with 5 points. With n=5, p values should be removed entirely as they imply inferential weight that this design cannot support. Weighted least squares add precision to each point but does not increase the actual sample size. With n=5 you can only describe these findings, you cannot generalize effect estimates. One option would be to reanalyze the data using provincial level data, instead of the five regions.
- Another weakness is that humidity is measured only for one month that is April 2022. DED is a chronic condition. Using one month of climate to explain lifetime prevalence is a severe weakness. Reading online I can see regions in Argentina have pronounced seasonal swings, April is relatively mild. Trying to correlate a single mild weather data point with a chronic condition does not make sense. If available the authors should incoprporate more months (ideally years or averages), or wind characteristics.
- Another problem that the authors acknowledge but underplay is that people spend most time indoors where humidity varies dramatically with HVAC use and doesn’t correlate with outdoor metoerorology. Thus the "humidity" variable is likely misaligned with actual ocular surface stress. This would usually bias toward the null, but here the association remains strong likely because of the tiny number of units (only 5).
- Sampling bias likely influences regional differences. The original dataset is voluntary and regional differences in internet acess, education, gender, healthcare access could easily artificially produce gradients. For instace, Cuyo has 85% women. Is this expected? From what I read it’s the opposite. This could alone shift DED prevalence.
- The Discussion should be adjusted completely as right now it drifts a lot toward implying causality, although this is only an ecological, cross sectional study.The present analysis could be only seen as hypothesis-generating, not confirmatory. Statements like “our findings support integrating climate into public-health strategies” are a serious overstatement. Or that “age/sex do not explain the association” are statistically underpowered and shouldn't be framed strongly.
Author Response
Thank you for the opportunity to review this manuscript.
Overall, this is a useful study with a clear message that is that lower relative humidity correlates with higher dry eye prevalence across Argentine regions. The study uses a very large sample size, ties the findings well to exisiting literature, writing is good and the visualizations are clean. However, there are several methodological weaknesses that limit the interpreatability and the strength of the conclusions. It stands good as a descriptive geographic report but it cannot support causal inferences whereas the authors overstate some associations.
RESPONSE: Dear reviewer #1, thank you very much for your comments and for taking the time to help us improve the scientific value of our work. All your comments have been taken into consideration, and we have made the necessary modifications. Below, we will respond to each point individually.
Specific comments:
COMMENT 1. The authors make correlations with n=5 regions, which are essentially uninterpretable statistically. The pearson r is meaningless with 5 points. With n=5, p values should be removed entirely as they imply inferential weight that this design cannot support. Weighted least squares add precision to each point but does not increase the actual sample size. With n=5 you can only describe these findings, you cannot generalize effect estimates. One option would be to reanalyze the data using provincial level data, instead of the five regions.
RESPONSE: We thank for this important methodological observation. We fully agree that ecological analyses based on only five regional units cannot support conventional statistical inference. In accordance with this comment, we have revised the manuscript to ensure that all region-level analyses are presented strictly as descriptive and exploratory, and without implying inferential strength.
Specifically: Removal of inferential statistics at the regional level; Pearson p-values, CIs, and model-based p-values have been removed from all region-level analyses; Correlation coefficients and regression outputs are now presented solely as descriptive summary measures, without inferential interpretation. Explicit clarification in the Methods: We added a detailed statement noting that, with n=5 regions, ecological correlations and weighted least-squares models are “mathematically estimable but not statistically stable,” and that results should be interpreted as exploratory patterns rather than inferential estimates. Also, we clarified that p-values apply only to non-ecological (group comparison) analyses. In the Discussion section, a new paragraph now emphasizes the limitations of ecological inference with n=5, the instability of parameter estimates, and the risk of ecological fallacy. Also, we state clearly that the climatic associations are descriptive patterns that require confirmation with seasonal or individual-level data.
Regarding the suggestion to re-analyze the data at the provincial level: We appreciate this valuable comment. While provincial-level DED prevalence is available, extending the ecological modelling to 23 provinces would not meaningfully strengthen the present analysis. The study was inherently constrained by the use of a single month of meteorological data (April 2022), and short-term outdoor humidity is unlikely to reflect long-term DED risk at a provincial scale. Under these conditions, province-by-province comparisons would add statistical granularity but not epidemiological validity, and could unintentionally amplify ecological error.
For this reason, the current work focuses on the five standard geopolitical regions, which accurately represent Argentina’s broad climatic zones and allow more coherent interpretation of environmental patterns. We have now clarified this rationale in the Methods. Importantly, we agree that finer geographic resolution will be essential in future research. One purpose of the present study is precisely to justify and inform the design of future multi-seasonal or multilevel studies capable of evaluating provincial variability with appropriate temporal alignment.
All corresponding revisions have been marked in the manuscript.
COMMENT 2. Another weakness is that humidity is measured only for one month that is April 2022. DED is a chronic condition. Using one month of climate to explain lifetime prevalence is a severe weakness. Reading online I can see regions in Argentina have pronounced seasonal swings, April is relatively mild. Trying to correlate a single mild weather data point with a chronic condition does not make sense. If available the authors should incoprporate more months (ideally years or averages), or wind characteristics.
RESPONSE: We thank the reviewer for this important observation. We fully agree that using climatic data from a single month (April 2022) represents a substantial limitation when studying a chronic condition such as DED, especially in a country with pronounced seasonal variability such as Argentina.
Unfortunately, expanding the climatic dataset is not feasible within the scope of the present study because this work constitutes a secondary analysis of an already completed nationwide cross-sectional survey, conducted exclusively in April 2022. As such, the study design inherently restricts us to meteorological conditions corresponding to the exact period in which data collection took place. Province- or region-level monthly exposures for other months cannot be linked to individual responses without introducing temporal misclassification.
To address this concern constructively, we have implemented the following revisions:
A. Strengthened Discussion on this limitation: we added explicit statements emphasizing that; April 2022 represents a single, mild, transitional month in Argentina; the use of one-month climatic data limits inferences regarding chronic or lifetime DED prevalence; exposure misclassification is probable, and the results should be interpreted as exploratory patterns, not causal estimates; seasonal variability and long-term climatology require future longitudinal or time-series studies.
B. Clarified in the Methods: we reinforce that climatic variables correspond strictly to April 2022 and that this constrains the ecological interpretation.
C. Forward-looking statement: ee agree with the reviewer that future research using: multi-month or multi-year averages, wind speed, UV radiation, pollutants, would offer stronger environmental characterization. We have added this point as a recommendation in the Discussion. Even with this limitation, presenting the April ecological snapshot remains valuable as a first exploration of regional climatic patterns, provided it is interpreted strictly as hypothesis-generating.
We appreciate this valuable feedback, which significantly improved the clarity and transparency of our manuscript.
COMMENT 3. Another problem that the authors acknowledge but underplay is that people spend most time indoors where humidity varies dramatically with HVAC use and doesn’t correlate with outdoor metoerorology. Thus the "humidity" variable is likely misaligned with actual ocular surface stress. This would usually bias toward the null, but here the association remains strong likely because of the tiny number of units (only 5).
RESPONSE: We thank the reviewer for raising this important point. We fully agree that outdoor humidity is an imperfect proxy for personal exposure, since individuals spend the majority of their time indoors, where relative humidity can differ substantially from outdoor measurements due to HVAC systems, building characteristics, and occupational environments. As correctly noted, this type of nondifferential exposure misclassification would typically bias climate–DED associations toward the null.
To address this concern, we have made the following revisions:
A. Expanded description in the Discussion: We strengthened the paragraph on exposure misclassification to explicitly highlight that: “outdoor meteorology does not necessarily reflect indoor conditions where individuals spend most of their time; this can lead to substantial measurement error, and any observed association must therefore be interpreted cautiously as an ecological signal rather than an estimate of individual-level exposure. We also explicitly state that this limitation may contribute to overlap in temperature plots and that the RH association may appear stronger partly because of the small number of ecological units (n = 5), which limits stability of effect estimates.
B. Clarification added to Methods: We added a sentence stating that climatic variables represent outdoor conditions only, and that indoor conditions could not be measured, which constrains the interpretation of humidity as an exposure variable.
C. Interpretative caution: We emphasize throughout the Discussion that the findings are descriptive and exploratory, and not intended as causal inferences at the individual level. These additions clarify the limitations of using outdoor humidity as a proxy for personal ocular surface exposure and ensure that the ecological nature of the study is appropriately contextualized.
We now explicitly acknowledge that the appearance of a strong ecological pattern may partly reflect the small number of units (n=5), further reinforcing the need for interpretative caution.
COMMENT 4. Sampling bias likely influences regional differences. The original dataset is voluntary and regional differences in internet acess, education, gender, healthcare access could easily artificially produce gradients. For instace, Cuyo has 85% women. Is this expected? From what I read it’s the opposite. This could alone shift DED prevalence.
RESPONSE: We agree that sampling bias is an important concern given the voluntary, online nature of the original survey. As noted, differential internet access, digital literacy, educational level, and healthcare engagement can vary across Argentine regions and may influence both participation rates and the demographic composition of respondents. These factors could indeed create artificial geographic gradients.
In particular, the reviewer correctly points out the disproportionate percentage of women in certain regions (e.g., 85% in Cuyo). This pattern is unlikely to reflect true population distributions and may be the result of differential response behavior—women are known to participate in health-related surveys more frequently, to have greater engagement with ocular surface symptoms, and to be more likely to report or seek care for DED. We have now explicitly acknowledged that such demographic imbalances may inflate symptom-based DED prevalence and contribute to apparent regional differences.
To address this concern, we have reinforced in the Discussion that regional contrasts must be interpreted with caution, as the observed patterns may reflect sampling artifacts rather than true epidemiological differences. We also highlight that online nonprobability sampling limits representativeness and that future nationwide studies should incorporate probabilistic sampling, active recruitment strategies, and weighting procedures to improve demographic balance.
Although these limitations constrain external validity, they do not invalidate the descriptive climate–DED patterns observed, which remain clearly hypothesis-generating rather than inferential.
These clarifications have now been added prominently in both the Abstract and the Discussion, as requested.
COMMENT 5. The Discussion should be adjusted completely as right now it drifts a lot toward implying causality, although this is only an ecological, cross sectional study. The present analysis could be only seen as hypothesis-generating, not confirmatory. Statements like “our findings support integrating climate into public-health strategies” are a serious overstatement. Or that “age/sex do not explain the association” are statistically underpowered and shouldn't be framed strongly.
RESPONSE: We thank the reviewer for this important clarification. We have substantially revised the Discussion to ensure that all interpretations remain strictly descriptive and explicitly hypothesis-generating. Statements that could imply causal inference or public-health recommendations have been softened. We now repeatedly emphasize that: the analysis is ecological and cross-sectional; the number of regions (n = 5) precludes any inferential or confirmatory conclusions; age/sex adjustments in ecological models cannot rule out confounding, and; the findings should not guide public-policy decisions without longitudinal or individual-level validation.
We have modified phrasing throughout the Discussion to reflect this, including removing or rewording statements related to public-health strategies and the interpretation of adjusted regional models. Revised text is highlighted in yellow in the manuscript.
Finally, dear Reviewer 1, on behalf of my co-authors, I would like to thank you for taking the time to evaluate and help us rethink and improve our study. In the new version, you will find several changes in the text, highlighted in yellow, where we have made modifications following both your suggestions and those of two other reviewers. We hope we have managed to answer all your questions. Thank you.
Reviewer 2 Report
Comments and Suggestions for Authors
Several methodological and editorial issues should be addressed before publication. Statistical interpretation is the primary concern, given the very small number of regional units (n = 5). A few textual errors and structural inconsistencies also deserve correction.
- Ecological modelling with five regions requires stronger caution: The manuscript presents Pearson correlations, weighted correlations, and weighted least-squares models using only five regional observations. Although the authors acknowledge that these analyses are “exploratory,” the presentation of p-values, confidence intervals, and R² values gives the impression of more robust inferential power than can be justified.
The text should explicitly and repeatedly clarify that, with n = 5, any regression model is mathematically estimable but not statistically stable; therefore, p-values should not be interpreted conventionally, confidence intervals are not reliable, and model fit statistics (e.g., R²) should be viewed as descriptive only. This caveat should be included in both the Methods and the Discussion sections.
- Sampling bias should be emphasised more prominently: The manuscript mentions that the survey was online, voluntary, and likely to over-represent “younger, more educated, urban women.” This is a substantial limitation because all regions show an unusually high proportion of women (80–86%), but digital access varies considerably across Argentine regions. Given that DED prevalence is strongly sex-related, uneven sampling could distort regional comparisons. This limitation should be described more clearly in the Abstract and further elaborated upon in the Discussion.
- More detail required on meteorological data collection: The manuscript states that meteorological data were curated per province using “principal stations,” but does not specify: How many stations were used per province? Whether the values are simple or weighted averages, whether elevation or rural/urban differences might affect humidity readings. Providing this information would enhance transparency and facilitate readers' interpretation of the ecological associations.
- Repetition and structural inconsistencies: There are a few issues that should be corrected:
In the Abstract, “DED prevalence (%) was estimated overall and by province/region” appears twice in consecutive sentences. Two different figures seem to be labelled “Figure 1”. Some wording and phrasing are duplicated in a way that interrupts the flow.
Author Response
Several methodological and editorial issues should be addressed before publication. Statistical interpretation is the primary concern, given the very small number of regional units (n = 5). A few textual errors and structural inconsistencies also deserve correction.
RESPONSE. Dear Reviewer 2, your comments have allowed us to review our data and the entire manuscript again. All of your comments have been practical and constructive. We have therefore modified the manuscript in accordance with your recommendations. Below you will find our responses, point by point.
COMMENT 1. Ecological modelling with five regions requires stronger caution: The manuscript presents Pearson correlations, weighted correlations, and weighted least-squares models using only five regional observations. Although the authors acknowledge that these analyses are “exploratory,” the presentation of p-values, confidence intervals, and R² values gives the impression of more robust inferential power than can be justified.
The text should explicitly and repeatedly clarify that, with n = 5, any regression model is mathematically estimable but not statistically stable; therefore, p-values should not be interpreted conventionally, confidence intervals are not reliable, and model fit statistics (e.g., R²) should be viewed as descriptive only. This caveat should be included in both the Methods and the Discussion sections.
RESPONSE: We thank the reviewer for this important methodological observation and fully agree that ecological analyses based on only five regional units cannot support conventional statistical inference. We appreciate the opportunity to clarify and strengthen the transparency of our analytic approach.
A. All p-values, confidence intervals, and significance statements related to ecological (regional) analyses have been removed. Weighted correlations and weighted least-squares (WLS) model outputs are now presented solely as descriptive summaries, without any inferential interpretation. P-values remain only for non-ecological comparisons, and this distinction is now explicitly stated in the Methods section.
B. Strengthened methodological caveats in the Methods section: we added explicit language noting that with n = 5 regions, ecological correlations and regressions are mathematically estimable but statistically unstable; Coefficients, R² values, and weighted correlations should be interpreted as descriptive patterns only; These analyses are intended exclusively for hypothesis generation and cannot be used for inferential testing.
C. Expanded caveats in the Discussion section: we incorporated a detailed paragraph emphasizing that: region-level results are susceptible to ecological fallacy; associations derived from five regional observations lack precision and cannot rule out confounding; climatic relationships must be confirmed through seasonal, longitudinal, or individual-level research designs.
D. Improved consistency and transparency throughout the manuscript. We revised terminology across the Results and Discussion to ensure that regional analyses are consistently described as exploratory, descriptive, and non-inferential, thereby avoiding any implication of confirmatory interpretation.
All revisions were incorporated as requested and are clearly indicated in the updated Methods and Discussion sections.
COMMENT 2. Sampling bias should be emphasised more prominently: The manuscript mentions that the survey was online, voluntary, and likely to over-represent “younger, more educated, urban women.” This is a substantial limitation because all regions show an unusually high proportion of women (80–86%), but digital access varies considerably across Argentine regions. Given that DED prevalence is strongly sex-related, uneven sampling could distort regional comparisons. This limitation should be described more clearly in the Abstract and further elaborated upon in the Discussion.
RESPONSE: We appreciate the reviewer’s observation regarding sampling bias, which is indeed one of the central limitations of the study. Dry eye disease is strongly sex-related, and the over-representation of women, combined with unequal digital access across regions, could distort regional prevalence estimates. We agree that this limitation requires more explicit emphasis.
Accordingly, we have implemented the following revisions:
- Abstract: We added a clear statement noting that the online, voluntary survey likely led to non-representative sampling, with overrepresentation of younger, more educated, urban women, which may influence regional comparisons. (The Abstract was revised to explicitly mention this limitation in the second sentence of the Methods section).
- Discussion: We expanded the sampling-bias paragraph to explicitly state that the high proportion of women (80–86% across regions) and regional differences in internet could distort geographic gradients in DED prevalence. We now highlight that these factors limit generalizability and must be considered when interpreting regional differences.
COMMENT 3. More detail required on meteorological data collection: The manuscript states that meteorological data were curated per province using “principal stations,” but does not specify: How many stations were used per province? Whether the values are simple or weighted averages, whether elevation or rural/urban differences might affect humidity readings. Providing this information would enhance transparency and facilitate readers' interpretation of the ecological associations.
RESPONSE. We thank the reviewer for requesting clarification on meteorological data collection, and we agree that additional detail improves transparency. In Argentina, the National Meteorological Service (SMN) operates approximately 125 surface observation stations distributed throughout the national territory. For the present study, climatic data (monthly mean temperature and relative humidity for April 2022) were extracted from the SMN stations located in each provincial capital, as these are the reference stations used for official climatological reporting in Argentina. This choice was also justified by the demographic structure of the country: approximately 90% of the population resides in urban areas, and provincial capitals contain the largest concentration of inhabitants within each province. For provinces with more than one SMN station, data from the principal synoptic station (the one designated by SMN as the official climatological reference) were used. The values reported by SMN for each station are simple monthly means, calculated from daily observations; no additional weighting was applied. Because the aim was to characterize broad regional climatic profiles rather than micro-environmental variability, we did not incorporate elevation adjustments or urban–rural weighting. We have now clarified these methodological details in the revised Methods section.
We fully acknowledge that using provincial-capital stations may not capture intra-provincial variability in humidity or temperature, and that these choices can introduce exposure misclassification. As noted in the Discussion, such misclassification is likely nondifferential and would bias ecological associations toward the null. We have reinforced this limitation in the revised manuscript.
COMMENT 4. Repetition and structural inconsistencies: There are a few issues that should be corrected: In the Abstract, “DED prevalence (%) was estimated overall and by province/region” appears twice in consecutive sentences. Two different figures seem to be labelled “Figure 1”. Some wording and phrasing are duplicated in a way that interrupts the flow.
RESPONSE. We thank the reviewer for highlighting these important clarity issues. In the revised manuscript, we have addressed all points as follows:
Abstract: the repeated sentence “DED prevalence (%) was estimated overall and by province/region” has been removed. The Abstract has been fully restructured to avoid redundancy, to follow a more concise flow, and to incorporate the reviewer’s suggestion regarding the need for stronger emphasis on sampling bias and the exploratory nature of regional analyses.
Figures: the labeling inconsistency in which two figures were both labeled “Figure 1” has been corrected. All figures are now consistently numbered and cross-referenced. Textual redundancy: phrasing that appeared duplicated in the Introduction, Methods, and Discussion has been edited for coherence and clarity. Sentences implying similar concepts were consolidated to improve scientific flow and eliminate unnecessary repetition.
Finally, dear reviewer 2, we appreciate all your comments. Each one has been considered and taken into account, and you will find the changes in the text highlighted in yellow. We hope that our new version has managed to resolve all the issues raised. Thank you again.
Reviewer 3 Report
Comments and Suggestions for Authors
Overall Assessment and Relevance
This manuscript examines the association between climatic variables—particularly relative humidity (RH) and temperature—and the prevalence of dry eye disease (DED) across Argentina. This is a highly relevant topic given the increasing recognition of environmental factors in ocular surface disease and the growing global interest in climate-related health determinants.
The work is publishable and potentially impactful, but several sections require substantial expansion, especially the literature contextualization, methodological clarifications, and the Discussion section.
Strengths
Large and geographically diverse sample
- Over 10,000 respondents from all regions, ensuring wide territorial representation.
Integration of meteorological data
- Monthly mean RH and temperature for April 2022 were correctly matched to provinces.
Clear data visualization
- Regional map illustrating DED prevalence
Major Issues and Recommendations
Introduction
The introduction provides a reasonable summary but remains too limited for an international journal.
Suggested expansions:
- Include more references from North America, Europe, and Asia studying climate and DED.
- Elaborate on the physiological impact of RH on tear film homeostasis.
- Add information about Argentina’s climatic heterogeneity (latitude, altitude, continentality, maritime influence).
Methods
Several clarifications are necessary:
- How meteorological stations were selected for each province (one station? multiple station averages?).
- Justification for using April 2022 data.
Results
Results are well presented, but you can strengthen the narrative:
- Emphasize that the region with the lowest RH (Cuyo, 55.1%) had the highest DED prevalence (49.8%).
- Highlight that the Northwest region had the highest mean age yet the lowest DED prevalence, adding evidence against an age confounding effect.
Discussion
This section needs the most expansion. Specific recommendations:
- Discuss contradictory evidence on temperature (as in your Figure 3).
- Explain possible protective effects of high RH in subtropical Northeast Argentina.
- Add implications for policy, urban design, and patient counseling.
Conclusions
The conclusions are appropriate but can be made stronger by adding:
- Clear statements about the robustness of the RH–DED association.
- Specific recommendations for future multi-seasonal studies.
- Implications for climate-informed ophthalmic practice.
Author Response
Overall Assessment and Relevance
This manuscript examines the association between climatic variables—particularly relative humidity (RH) and temperature—and the prevalence of dry eye disease (DED) across Argentina. This is a highly relevant topic given the increasing recognition of environmental factors in ocular surface disease and the growing global interest in climate-related health determinants.
The work is publishable and potentially impactful, but several sections require substantial expansion, especially the literature contextualization, methodological clarifications, and the Discussion section.
Strengths
Large and geographically diverse sample: Over 10,000 respondents from all regions, ensuring wide territorial representation.
Integration of meteorological data: Monthly mean RH and temperature for April 2022 were correctly matched to provinces.
Clear data visualization: Regional map illustrating DED prevalence
RESPONSE. Dear Reviewer 3#, thank you for your time and kind words. This really encourages us to improve. We have taken all your comments on board and you will find our responses below. Thank you again.
Major Issues and Recommendations
COMMENT 1. Introduction
The introduction provides a reasonable summary but remains too limited for an international journal.
Suggested expansions:
-
Include more references from North America, Europe, and Asia studying climate and DED.
-
Elaborate on the physiological impact of RH on tear film homeostasis.
-
Add information about Argentina’s climatic heterogeneity (latitude, altitude, continentality, maritime influence).
RESPONSE. We thank the reviewer for this valuable suggestion. In the revised version, we have substantially expanded the Introduction to provide broader international context and a more detailed description of the physiological relevance of environmental factors. Specifically, we incorporated seven additional references from North America, Europe, and Asia addressing climate–DED relationships, tear-film physiology under varying humidity conditions, and global epidemiological patterns.
We also expanded the description of Argentina’s climatic heterogeneity, including latitude, altitude, and continentality gradients, to better contextualize regional environmental differences. These additions strengthen the scientific background and improve the international relevance of the manuscript.
COMMENT 2. Methods
Several clarifications are necessary:
-
How meteorological stations were selected for each province (one station? multiple station averages?).
-
Justification for using April 2022 data.
RESPONSE. We thank the reviewer for these constructive suggestions. Both points have now been fully clarified in the revised Methods section.
Selection of meteorological stations. We have added a detailed description explaining that meteorological data were obtained from the National Meteorological Service and that, for each province, we used the principal station located in the provincial capital. This approach ensures standardized reporting and reflects population-dense areas, where most survey respondents reside.
Justification for using April 2022 data. We explicitly state that April 2022 was the only month during which the nationwide survey was conducted, making it the only period that could be meaningfully linked to climatic exposures. We also clarify that April represents a climatically mild transition month in Argentina, minimizing extremes and allowing regional ecogeographical differences to be evaluated more cleanly.
COMMENT 3. Results
Results are well presented, but you can strengthen the narrative:
-
Emphasize that the region with the lowest RH (Cuyo, 55.1%) had the highest DED prevalence (49.8%).
-
Highlight that the Northwest region had the highest mean age yet the lowest DED prevalence, adding evidence against an age confounding effect.
RESPONSE. We thank the reviewer for this helpful suggestion. In the revised manuscript, we have strengthened the Results narrative by explicitly highlighting these descriptive contrasts. We now state that Cuyo—the region with the lowest relative humidity (55.1%)—had the highest DED prevalence (49.8%), and that the Northwest, despite having the oldest age profile, showed the lowest prevalence. These additions appear in Sections 3.1 and 3.2 and improve the clarity of the geographic and climatic patterns observed.
COMMENT 4. Discussion
This section needs the most expansion. Specific recommendations:
-
Discuss contradictory evidence on temperature (as in your Figure 3).
-
Explain possible protective effects of high RH in subtropical Northeast Argentina.
-
Add implications for policy, urban design, and patient counseling.
RESPONSE. We thank the reviewer for these valuable suggestions. The Discussion section has been substantially expanded. We now (i) address the contradictory evidence regarding temperature effects and relate this to the absence of a temperature gradient in our regional data, (ii) elaborate on potential protective effects of high relative humidity in the subtropical Northeast of Argentina, including mechanisms related to reduced evaporative stress and tear-film preservation, and (iii) include a more cautious, explicitly non-causal framing of how future research may inform public-health guidance, urban planning, and patient counseling. All additions are highlighted in the revised Discussion.
COMMENT 5. Conclusions
The conclusions are appropriate but can be made stronger by adding:
-
Clear statements about the robustness of the RH–DED association.
-
Specific recommendations for future multi-seasonal studies.
-
Implications for climate-informed ophthalmic practice.
RESPONSE. We thank the reviewer for this constructive suggestion. In the revised manuscript, the Conclusions section has been substantially strengthened to address all three points. Specifically:
We now describe the RH–DED association as descriptively robust, while explicitly noting the ecological nature and inferential limitations of the five-region design.
We added specific recommendations for future research, including multi-seasonal, longitudinal, and individual-level approaches to capture temporal variability and finer-scale environmental exposures.
We incorporated a brief and appropriately cautious statement on the potential implications for climate-informed ophthalmic practice, emphasizing that such applications remain speculative until confirmed by future studies.
These revisions enhance the clarity, balance, and translational relevance of the Conclusions section. The updated text is included in the manuscript (Section 5, Conclusions).
Finally, reviewer 3, you will find all the modifications in the text highlighted in yellow. These reflect the changes we have made based on your valuable comments, as well as the modifications made at the request of the other reviewers.
We hope you like our new and improved version. Thank you again.
Round 2
Reviewer 1 Report
Comments and Suggestions for Authors
Most of my earlier concerns have been addressed. A few scientific points remain:
- The abstract and main text still call this a “nationwide population dataset,” but the methods describe a voluntary online survey. Please describe it consistently as a nationwide online / non‑probability sample and state clearly that prevalence and regional differences are not population‑representative, but reflect this specific group.
- In the Results and Figure 1, Patagonia is said to have the highest rate as “cases per 100,000 inhabitants.” Since the numerator comes from a voluntary survey, these numbers are essentially survey prevalence projected onto census counts, not true population rates. I suggest either making this explicit (estimated values based on survey prevalence) or avoiding the “per 100,000” phrasing.
- The ecological models adjust for age and sex, but other regional factors likely linked to both DED and humidity (urbanization, jobs, screen use, contact lens wear, healthcare access, pollution, etc.) are still not included. Please state in the Discussion that regional RH may reflect several unmeasured behavioural and environmental factors, so the pattern cannot be attributed to humidity alone.
- DED is defined as severe symptoms or prior ophthalmologist‑confirmed diagnosis. Because access to eye care and probability of getting a formal diagnosis vary by region, the “prior diagnosis” part of the definition may create higher apparent prevalence in areas with better access and awareness, independent of true disease burden. It would be useful to mention that this mixed definition can bias regional prevalence upward in better‑resourced regions, and that including severe symptoms only partly reduces this issue.
- Climatic exposure is based on monthly mean RH and temperature in April 2022. Short‑term episodes of very low RH, strong wind, or temperature extremes are probably more relevant for the ocular surface than monthly means, and capital‑city stations may not capture conditions across large, heterogeneous provinces. Please state more explicitly that monthly means may hide important extremes and capital‑based data may not reflect within‑province variability, so the exposure metric is just a coarse ecological proxy.
- The Discussion and Conclusions correctly describe the findings as hypothesis‑generating and non‑causal. However, some sentences still suggest that regional RH “contributes” to differences in DED. Clarifying this would make the interpretation fully consistent with the limitations.
- Since you report that DED prevalence was estimated at both provincial and regional levels, a supplementary table with provincial prevalence and April RH/temperature (descriptive only) would be helpful.
Author Response
- 1. The abstract and main text still call this a “nationwide population dataset,” but the methods describe a voluntary online survey. Please describe it consistently as a nationwide online / non‑probability sample and state clearly that prevalence and regional differences are not population‑representative, but reflect this specific group.
RESPONSE:
Dear reviewer, thank you
very much for your comments. In our new version, we have made all the corrections you mentioned. Below, we will respond to each one individually.
We agree and have revised the manuscript to consistently describe the dataset as a nationwide, voluntary online survey (non-probability sample). We now explicitly state that no post-stratification weighting to census margins was applied and that estimates and regional contrasts are sample-based, not population-representative. Specific edits are highlighted in light blue.
- 2. In the Results and Figure 1, Patagonia is said to have the highest rate as “cases per 100,000 inhabitants.” Since the numerator comes from a voluntary survey, these numbers are essentially survey prevalence projected onto census counts, not true population rates. I suggest either making this explicit (estimated values based on survey prevalence) or avoiding the “per 100,000” phrasing.
RESPONSE:
We agree. We have removed the “per 100,000 inhabitants” phrasing throughout and now report prevalence (%) only. Where relevant, we explicitly state that estimates reflect the survey sample (non-probability).
- 3. The ecological models adjust for age and sex, but other regional factors likely linked to both DED and humidity (urbanization, jobs, screen use, contact lens wear, healthcare access, pollution, etc.) are still not included. Please state in the Discussion that regional RH may reflect several unmeasured behavioural and environmental factors, so the pattern cannot be attributed to humidity alone.
RESPONSE:
Thank you. Now, we have added this information in the Discussion, with a new paragraph, acknowledging potential residual confounding and noting that regional RH may capture unmeasured behavioural and environmental factors; therefore, the observed pattern cannot be attributed to humidity alone. We also point to the need for multilevel designs with richer covariate data.
- 4. DED is defined as severe symptoms or prior ophthalmologist‑confirmed diagnosis. Because access to eye care and probability of getting a formal diagnosis vary by region, the “prior diagnosis” part of the definition may create higher apparent prevalence in areas with better access and awareness, independent of true disease burden. It would be useful to mention that this mixed definition can bias regional prevalence upward in better‑resourced regions, and that including severe symptoms only partly reduces this issue.
RESPONSE:
We have added information clarifying that our mixed case definition may introduce ascertainment bias across regions with differing access to care and awareness. We note that a symptoms-only definition can reduce but not eliminate this issue (e.g., residual differences in health-seeking behavior and digital literacy). The Discussion has been updated accordingly.
- 5. Climatic exposure is based on monthly mean RH and temperature in April 2022. Short‑term episodes of very low RH, strong wind, or temperature extremes are probably more relevant for the ocular surface than monthly means, and capital‑city stations may not capture conditions across large, heterogeneous provinces. Please state more explicitly that monthly means may hide important extremes and capital‑based data may not reflect within‑province variability, so the exposure metric is just a coarse ecological proxy.
RESPONSE:
We have clarified in the Discussion that monthly means can obscure short-term extremes (e.g., very low RH, strong wind, temperature spikes) and that capital-based measurements may not represent conditions across large, heterogeneous provinces. We now state explicitly that our exposure metric is a coarse ecological proxy. We also note this in Methods as a measurement-level limitation.
- 6. The Discussion and Conclusions correctly describe the findings as hypothesis‑generating and non‑causal. However, some sentences still suggest that regional RH “contributes” to differences in DED. Clarifying this would make the interpretation fully consistent with the limitations.
RESPONSE:
Thank you. Now, the manuscript was revised to avoid causal wording. We now use non-causal phrasing and explicitly frame results as hypothesis-generating.
- 7. Since you report that DED prevalence was estimated at both provincial and regional levels, a supplementary table with provincial prevalence and April RH/temperature (descriptive only) would be helpful.
RESPONSE:
Dear reviewer, we appreciate the suggestion. In exploratory work, we did compute provincial proportions; however, given the non-probability sampling, uneven provincial sample sizes, and capital-station exposure assignment, several province-level estimates were unstable and at higher risk of misinterpretation. To reduce noise and ecological misclassification we therefore focused the presentation on regional summaries, which better reflect broad climatic patterns and are the level at which our ecological analyses were prespecified. At this time we are not in a position to assemble a reliable province-level supplementary table. We have added a note in Methods to clarify this reporting choice. We would, of course, be happy to revisit province-level reporting in future work specifically designed (and powered) for that granularity.
In summary,
Dear reviewer, we are very grateful for all your comments, which have allowed us to review and improve many fundamental aspects of our work. But also, as is often the case in research, it is allowing us to think about how to improve a future study, both for our country, Argentina, and for neighboring countries. Thank you very much indeed. We hope that this work will be published, as this will justify our future studies. Once again, thank you very much.